# Learning Carbohydrate Digestion and Insulin Absorption Curves Using Blood Glucose Level Prediction and Deep Learning Models

**DOI:** 10.3390/s21144926

**Published:** 2021-07-20

**Authors:** Mario Muñoz-Organero, Paula Queipo-Álvarez, Boni García Gutiérrez

**Affiliations:** 1Department of Telematic Engineering, Universidad Carlos III de Madrid, Leganés, 28911 Madrid, Spain; pqueipo@it.uc3m.es (P.Q.-Á.); bogarcia@it.uc3m.es (B.G.G.); 2UC3M-BS Institute of Financial Big Data, Universidad Carlos III de Madrid, Leganés, 28911 Madrid, Spain

**Keywords:** type 1 diabetics, glucose estimation, long short-term memory (LSTM), deep learning, insulin absorption, carbohydrate digestion, artificial intelligence

## Abstract

Type 1 diabetes is a chronic disease caused by the inability of the pancreas to produce insulin. Patients suffering type 1 diabetes depend on the appropriate estimation of the units of insulin they have to use in order to keep blood glucose levels in range (considering the calories taken and the physical exercise carried out). In recent years, machine learning models have been developed in order to help type 1 diabetes patients with their blood glucose control. These models tend to receive the insulin units used and the carbohydrate taken as inputs and generate optimal estimations for future blood glucose levels over a prediction horizon. The body glucose kinetics is a complex user-dependent process, and learning patient-specific blood glucose patterns from insulin units and carbohydrate content is a difficult task even for deep learning-based models. This paper proposes a novel mechanism to increase the accuracy of blood glucose predictions from deep learning models based on the estimation of carbohydrate digestion and insulin absorption curves for a particular patient. This manuscript proposes a method to estimate absorption curves by using a simplified model with two parameters which are fitted to each patient by using a genetic algorithm. Using simulated data, the results show the ability of the proposed model to estimate absorption curves with mean absolute errors below 0.1 for normalized fast insulin curves having a maximum value of 1 unit.

## 1. Introduction

Type 1 diabetes is among the most frequent chronic diseases in children and young adults and has an important social and health impact that affects the quality of life of the people who suffer from it. The disease begins when the pancreas does not produce enough insulin or when it is not able to use the insulin produced efficiently. As a result, an increase in the level of glucose in the blood can cause heart and kidney problems or damage some of the nerves in the body.

Despite the latest advances, an exact mechanism for curing the disease is not known. This increases the importance of proposing and validating techniques that facilitate the monitoring and treatment of type 1 diabetics [1] and even the development of systems that allow the identification of patterns that can help to find the best treatment for each patient.

Machine learning techniques and algorithms are proving to be able to build a solid foundation in many areas in general and for blood glucose prediction in particular [2,3]. Data from real patients, by applying data-based models and techniques able to learn and extract patterns from the data, have been used for monitoring patients and predicting possible drops or rises in blood glucose levels, establishing recommendations based on each user’s particular and personal data. 

Currently, both physiological inspired models [4] and machine learning models (in particular deep learning-based models [3]) are being successfully applied to data obtained from blood glucose monitors (Continuous Glucose Monitors or CGM) together with manually provided information such as insulin boluses and meals. Physiological-inspired models are mapped to differential equations controlling the blood glucose level kinetics based on user-dependent parameters, which are sometimes hard to estimate based on the data collected from user glucose sensors (such as CGM) and insulin and carbohydrate intakes. Machine learning models represent black-box generic models with some internal parameters that are tuned to fit the data obtained for the sensors in order to minimize the error when using the model for estimating unknown values. Machine learning models try to find patterns in blood glucose levels based on the times, types, and amounts of insulin used and the carbohydrate content taken. Physiological-inspired models [4] show that the human body follows a process that takes an absorption step (both for carbohydrates and insulin) first and then uses the plasma insulin and glucose levels to guide the different insulin and glucose-dependent processes in the human organs and tissues. Intuitively, applying machine learning models directly to insulin bolus units consumed and carbohydrate content in meals taken will make it more difficult to learn blood glucose patterns than using carbohydrate and insulin absorption curves. However, carbohydrate and insulin absorption curves can not be directly measured from the glucose sensors used by type 1 diabetes patients.

This paper proposes a novel mechanism to estimate patient-dependent meal digestion and insulin absorption curves based on data-driven genetic optimization and uses the estimated curves in order to optimize the performance of a Recurrent Neural Network (RNN) based on Long Short-Term Memory (LSTM) cells in order to estimate future levels of blood glucose.

## 2. State of the Art

Type 1 diabetes mellitus (T1DM) patients adjust insulin doses in order to properly fit the carbohydrate intake to keep blood glucose levels in range. An optimal regulation of blood glucose levels will require a proper combination of timely injections of insulin boli (considering the number of insulin units and the type of insulin to be used) and food intake depending on the current and past blood glucose levels. 

Several absorption models for insulin and carbohydrates have been proposed in order to assess the influence over time of human actions (meal intake or insulin injections) into plasma concentration levels and therefore provide an estimation of the impact on blood glucose levels. Chiara et al. [5] proposed a simulation model in normal humans that describes the physiological events that occur after a meal. The parameters for the model were set to fit the mean data of a large normal subject database that underwent a triple tracer meal protocol that provided quasi-model-independent estimates of major glucose and insulin fluxes. A parametric model was developed decomposing the system into subsystems. Wilinska et al [6] proposed a model for insulin Lispro kinetics with bolus and Continuous Subcutaneous Insulin IInfusion (CSII) modes of insulin delivery. Eleven alternative models of insulin kinetics were proposed. The models considered several absorption approximations, taking into account different rates of absorption and considering the influence of the volumes of insulin. The models also studied two pathways for insulin absorption (fast and slow) and were based on compartment models. 

Absorption models for insulin and carbohydrates can be expressed in mathematical terms in the form of differential equations. A particular mathematical representation can be found in Ruan et al [4]. The authors developed a new hierarchical model to relate subcutaneous insulin delivery and carbohydrate intake to continuous glucose monitoring over 12 weeks while describing day-to-day variability. A compartment model that comprised five linear differential equations was proposed. Hovorka et al. [7] also proposed a compartment model, which represented the glucoregulatory system and included submodels representing the absorption of subcutaneously administered short-acting insulin Lispro and gut absorption, showing promising results. Hajizadeh et al. [8] presented a Plasma Insulin Concentration (PIC) estimator that incorporated Hovorka’s glucose–insulin model [7] with the unscented Kalman filtering algorithm. The authors took an interest in improving convergence. Haiya et al. [9] modeled the insulin therapies using a delay differential equation model. The authors studied the dynamics of the model both qualitatively and quantitatively. Lehmann and Deutsch [10] implemented the equations in the model in [11] for in silico experimentations in the AIDA2 simulator [12]. The authors used numerical integration in order to find solutions to the differential equations guiding the glucose kinetics in a simulated patient. The model contains a single glucose pool representing extracellular glucose (including blood glucose) into which glucose enters via both intestinal absorption and hepatic glucose production [10]. Intestinal absorption is modeled using a first-order equation. Glucose enters the portal circulation using this absorption model from the gut. Insulin absorption is also guided by a differential equation in which the change in plasma insulin concentration is controlled by first-order rate constant elimination and an insulin absorption model that considers the delay in action [10]. 

Machine learning techniques have also been used as an alternative family of methods in order to estimate upcoming values of blood glucose (BG) levels. Machine learning methods are able to learn from the experience (using labeled training data) how to combine the information measured from glucose, carbohydrate, and insulin signals in order to minimize the errors in the prediction of upcoming blood glucose (BG) levels. Machine learning models are based on multiple parameters that can be trained in order to approximate or simulate the underlying physiological model. However, generic machine learning models are not physiologically related to the human body glucose kinetic processes. 

Different machine learning models have been proposed in order to predict future Blood glucose (BG) levels are based on the knowledge of factors such as insulin dosage, nutritional intake, physical exercise, and recent BG levels. Pappada et al. [13] proposed the use of a Neural Network-based model to determine the impact of insulin, meals, daily activities, lifestyle, and emotional states on predicting glycemic trends. The study found that the Neural Network model worked well at predicting normal and hyperglycemic events but provided worse results for hypoglycemia episodes. A different machine learning model based on the implementation of a Support Vector Regression (SVR) algorithm that used a Kalman filter in order to estimate hidden values in the model was proposed in [14]. The model was trained for blood glucose estimation based on current and past levels of carbohydrates, insulin, and BG levels. The authors found that the prediction results provided by the machine learning model were comparable to those manually predicted by a doctor. Another study that uses a machine learning model based on SVR to estimate upcoming BG levels can be found in [15]. The authors limited the input signal to the information provided by a Continuous Glucose Monitoring (CGM) device. The study added a Differential Evolution (DE) algorithm to improve results. A different model based on Artificial Neural Networks (ANNs) for predicting blood glucose levels can be found in [16]. The study also focused on the use of CGM data alone. A different approach for BG estimations using a meta-learning approach can be found in [17]. The results are based on the use of regularized learning algorithms. The paper designed a mechanism that facilitated the portability of the model from patient to patient.

Recently, some deep learning algorithms have also been applied to the prediction of BG levels trying to achieve a better performance compared to previously used methods. The authors in [18] showed some methods for deep multi-output blood glucose forecasting and validated that the results using deep learning methods outperformed previous shallow learning alternatives. Mhaskar et al. [19] also proposed a deep learning approach to BG level estimation based on the previous BG levels but using a pre-clustering mechanism to train specific models for hypo, eu and hyperglycemic segments. The authors also demonstrated that deep learning methods can outperform shallow networks.

In this manuscript, we propose and validate a new mechanism that combines some information from the glucose kinetic physiological processes with the ability of deep machine learning models to adapt to each individual data in order to achieve better glucose level predictions. The model incorporates physiological absorption models for insulin and carbohydrates into the pre-processing steps of a machine learning algorithm using a Recurrent Neural Network based on LSTM cells. Then, the learning algorithm is able to learn the intricate dependencies between blood sugar levels and estimated values for blood insulin and carbohydrate concentrations trying to make it simpler for the deep learning model to learn. 

## 3. Materials and Methods

This section describes the dataset that has been used in order to evaluate the proposed model. The section also contains the details of the model and the methods used for validation.

### 3.1. Dataset Used

The AIDA2 simulator [10] is based on an in silico model for the glucose–insulin interaction in the human body. The simulator comes with 40 different case scenarios, which can be extended. The simulator generates blood glucose curves as well as plasma insulin curves based on the amount of carbohydrates and insulin doses taken. The AIDA2 simulator uses numerical integration in order to solve the differential equations guiding the glucose kinetics inside a human body [10]. The model is based on 4 differential equations along with 12 auxiliary relations, which are captured in [10]. The derivative of the insulin concentration over time has a positive term proportional to the rate of insulin absorption divided by the volume of insulin and a negative term controlled by the rate of insulin elimination [10]. The change of glucose concentration with time is also guided by a differential equation that takes into account the systemic appearance of glucose via glucose absorption from the gut, the overall rate of peripheral and insulin-independent glucose utilization, the net hepatic glucose balance, the rate of renal glucose excretion, and the volume of distribution for glucose [10].

In this manuscript, we want to analyze if meal digestion and insulin absorption curves can be learned by a proposed machine learning model using genetic optimization. The AIDA2 simulator constitutes a convenient dataset, since it makes the information about insulin absorption curves computed by the simulating model available, which can be used in order to validate the approach proposed in this manuscript. The AIDA2 simulator [12] has also been used in previous research studies such as [20,21] in order to validate blood glucose prediction algorithms. Figure 1 shows the result for a particular simulated patient for the plasma insulin levels from a single simulation in which the levels of plasma insulin are derived from the amounts and types of insulin doses taken (shown in the lower part of the figure). The change in plasma insulin depends on the rate of insulin absorption, the volume of insulin distribution, and a first-order rate constant for insulin elimination. The numerical integration of the model shows plasma insulin curves that have a growing first part in which the insulin absorption is higher than the insulin elimination, and a second decreasing part is controlled by the dominant elimination part.

The characteristics of the dataset used are captured in Table 1. For each case scenario provided by the AIDA 2 simulator [10], a total of 8 days of data is generated. The average number of insulin injections per day has been set to 4, and the average number of meals per day has been set to 6. 

### 3.2. Proposed Model

Deep learning models for blood glucose (BG) prediction previously found in the literature such as [18] use the raw information from insulin and carbohydrate intakes directly as inputs to a multilayer model, which tries to estimate the BG levels in a prediction horizon (PH) (normally of 30 or 60 min). The model we propose uses a two-step approach in which insulin and carbohydrate absorption curves are estimated first and then used to feed a multilayer deep learning model. 

Insulin and carbohydrate absorption curves can be estimated by solving mathematical models characterizing the glucose metabolism such as [11]. The software in [12] has used numerical integration to solve the differential equations in [11]. Solving the models for each user based on available measured data from Continuous Glucose Monitors (CGM) requires the estimation of internal parameters for each mathematical model, which could be difficult in some cases. In order to simplify the machine learning process to fit the available data, we propose an absorption curve generation process that will use a linear approximation for the first ascending part of the curve as used in [9] and an exponential decay for the second descending part of the curve as in [22].

A deep learning Recurrent Neural Network (RNN) model based on LSTM cells [19] will be used in order to generate estimates for the upcoming levels for blood glucose (BG) based on current and past levels for the same BG signal and the output of the 3 curve generation processes, one for the digestion of carbohydrates and the other two for fast and slow insulin absorption. The curve generators will receive as parameters the time to peak and the after-peak decay rate in order to generate signals, which will approximate the human body absorption processes. The model is shown in Figure 2. The details for the RNN part of the model are captured in Table 2. 

The AIDA2 simulator provides data samples every 15 min. In order to feed the model in Figure 2, the input data are pre-processed in order to generate 4-h windows of data for each of the input signals. No further pre-processing steps have been required, since the data contain no measurement errors. The output of the model in Figure 2 will be the estimated level for blood glucose in a prediction horizon (PH) of 60 min. The RNN part of the model in Figure 2 will receive the generated absorption curves together with the values measured from the CGM device as the input information in order to learn patterns in the data and fit the weights to minimize the output estimation error. The RNN model is expected to be able to reduce the estimated errors for blood glucose levels in the PH when the generated input signals optimally fit the real carbohydrate and insulin blood concentrations, since these values are the ones used in the internal model used to generate the real output data.

In order to optimally learn the absorption curves for carbohydrates and slow and fast insulin, the RNN model in Figure 2 will be trained using the output for each of the 3 curve generator processes using different values of peak times and decay rates. In order to generate the 3 absorption curves, a total of 6 parameters (3 peak times and 3 decay rates) have to be set. The RNN part of the model in Figure 2 will then be trained, and the output error after training will be used as an accuracy value for validating the similarity between the generated curves and the real ones. A genetic algorithm is used in order to find the optimal values for the 6 parameters controlling the generation of the absorption curves. In particular, the algorithm described in chapter 7 in reference [23] is used. The algorithm concatenates 4 major functions in order to find the individuals that optimize the value of a fitness function: recombine, mutate, evaluate, and select. The recombine function generates individuals that combine the attributes from those in the previous generation in order to find new individuals. The mutate function makes some random changes with a particular probability to the attributes in each individual. The evaluate function calculates the fitness values for all the individuals in a population. Finally, the selection function chooses the best individuals in order to generate a new population. The algorithm can iterate several times or epochs. An initial population of 15 individuals has been used, including a mutation probability of 20% in order to generate 15 generations of individuals. The best-fitted individuals have been selected for each generation. The process is shown in Figure 3. The initial values for each individual have been randomly selected. The maximum and minimum values for peak times have been set from 0 to 2 h for meal digestion and fast insulin concentration and from 0 to 12 h for slow insulin absorption. 

## 4. Experimental Results

### 4.1. Software Configuration

The model in Figure 2 and Figure 3 has been implemented in Python. The DEAP library [24] has been used in order to implement the genetic algorithm-based optimization process for estimating the optimal parameters used to generate the absorption curves. Each individual in the population is defined by the values for the six parameters controlling the generation of the curves for insulin and carbohydrate absorption processes (peak times and decay rates). A FitnessMax function has been used with weights equals to -1 so that a minimization problem is solved. An initial population of 15 individuals has been selected. The model in Figure 2 is used as the fitness function so that the individuals able to generate curves which are able to better train the RNN model are preferred. The cxTwoPoint parameter is used to cross individuals. The mutGaussian parameter is used to define the mutation of the individuals. A mutation probability of 20% has been selected.

The parameters for the LSTM-based Recurrent Neural Network in Figure 2 are captured in Figure 4. Two LSTM layers are stacked in order to augment the capacity of the model to learn intricate patterns and dependencies from the input data. Two-layer models have been previously used in studies such as [19], showing good results. The number of memory units in the LSTM cells for the first layer has been set to 10 and has been reduced to 5 in the second LSTM layer. These values have been previously validated in studies such as [21]. Dropout regularization layers have also been used in order to minimize overfitting problems. Two dense layers are connected at the output of the second LSTM layer in order to generate a single prediction value to estimate the level of blood glucose in the prediction horizon of 60 min. A Keras sequential model [25] has been used in order to implement the model in Figure 4. 

### 4.2. Numerical Results 

The AIDA2 simulator [10] has been used to simulate data for 8 consecutive days for 40 different reference T1DM patients. The model in Figure 2 and Figure 3 has been used in order to estimate the absorption curves for each particular patient. The parameters defining the differential equations implemented by the AIDA2 simulator [10] for each patient consider different characteristics that try to accommodate real cases for different users. Therefore, the model in Figure 2 and Figure 3 has been used for each patient independently. The AIDA2 simulator [10] provides access not only to the time evolution of blood glucose levels but also the plasma insulin concentration levels over time. We have used the insulin curves provided by the simulator as the desired or optimal curves to feed the RNN network in our model, since these curves are the real ones used by the simulator in order to calculate future blood glucose (BG) levels. The mean absolute error between the plasma insulin curve provided by the simulator and the approximated curve learned by the proposed model has been computed for each simulated use case. The average results for the learned insulin curves are captured in Table 3. The graphical representation for the AIDA2 provided and the learned curves are shown in Figure 5. Both curves have been normalized so that their maximum value is 1. The results in Figure 5 show that the genetic algorithm obtains a generated insulin curve that has a peak 90 min after the bolus injection and the decay rate is also learned to mimic the behavior of the curve used by the simulator (Figure 5 shows that both curves coincide around 3 and a half h after the bolus injection).

The genetic algorithm converges after 10 iterations (as shown in Figure 6). Each individual in the population contains the peak values and decay rates, which are used to generate the insulin and carbohydrate absorption curves, which together with the past blood glucose (BG) values are used as the input data to feed the RNN model in Figure 2. The input data are divided into training and validation subsets (70%/30%). The training samples are used to train the RNN model in Figure 2. The validation subset is used to compute the output value for the fitness function that the genetic algorithm uses in the optimization process. The initial population (iteration 0 in Figure 6) is composed of randomly generated values for the input parameters in order to generate random initial absorption curves. After training the RNN in Figure 2 with the different generated curves in the initial population, a significant difference is observed among the different prediction errors (mean value of 1, min value of 0.55, and max value of 2.45). Therefore, the shapes of input curves are important in order for the RNN to be able to predict upcoming values of blood glucose levels based on the past values of insulin and carbohydrate intakes. The best individuals are crossed, and a 20% mutation is introduced in order to generate a new generation of individuals for the next epoch in the optimization process performed by the genetic algorithm. Figure 6 captures the output values for the population used in each of the epochs for one particular patient generated using the AIDA2 simulator. After 10 iterations, the genetic algorithm is not able to significantly improve the accuracy achieved by the RNN in the model. 

The patient in Figure 5 and Figure 6 uses a combination of slow and fast-acting insulin. In order to assess the fast insulin curve learning process minimizing the effect of other types of insulin in the data, a patient using only fast-acting insulin has been simulated. The results are captured in Figure 7. Bolus injection takes place at instant 0 in the curve. The linear approximation for the first part of the curve follows the real curve, achieving a maximum value at the same time. The decay rate in this case is a bit faster than the real curve used by the simulator.

Complex machine learning models using several LSTM layers have shown better performance when predicting upcoming values of blood glucose (BG) levels than shallow models [3,18,19,21]. A good machine learning model will be able to compensate some inaccuracy levels in the input data. In order to assess the influence of the use of simpler machine learning models for the performance of the genetic algorithm in finding absorption curves that try to fit the real curves used by the AIDA2 simulator, an RNN model with a single LSTM layer with only five memory cells has been used. The results are captured in the next subsection. 

### 4.3. Using a Simple Model Results 

The model in Figure 8 consisting of an RNN with a single LSTM layer has been used instead of the model in Figure 4 in order to assess the influence of the machine learning model when trying to find optimal absorption curves. Previous studies such as [3,18,19,21] have shown that deeper machine learning models are able to increase the accuracy of blood glucose level predictions, both for horizons of 30 and 60 min, compared to models with a single LSTM layer such as the one in Figure 8. However, the use of a simpler machine learning model with a fewer number of parameters to tune will require less memory and training time and will therefore reduce the time needed to find the optimal absorption curves. The model in Figure 8 uses a single LSTM layer with only five memory cells and a final single neuron output layer to generate the blood glucose level prediction. A dropout layer is added as a regularization layer to the model.

The results in Figure 9 show the validation error for the first 10 epochs for the same patient generated by the AIDA2 simulator previously shown in Figure 5. The genetic algorithm is able to converge in 10 epochs using the simpler model. However, the average validation error in this case is 0.29 mmol/L (5.22 mg/dL), which is significantly higher than the validation error of 0.095 mmol/L (1.71 mg/dL) achieved by the two-layer model presented in Section 4.1. This result confirms previous results in studies such as [3,18,19,21] in which the impact of the model for predicting upcoming values of blood glucose levels is assessed. Comparing the prediction results with those in Table 4 in reference [21], the addition of absorption curves fitted to each patient, as proposed in this paper, is able to improve the Root Mean Square Error (RMSE) when using similar LSTM-based deep learning models. The authors in [21] achieved an RMSE value of 3.45 mg/dL using the same dataset as in this paper but only for a prediction horizon of 30 min and an optimized deep learning model. Using the two-layer model in Section 4.1, the absorption curves are able to improve the RMSE value to 1.71 mg/dL for a prediction horizon of 60 min.

In order to estimate the quality of the absorption curves found by the genetic algorithm when using the simpler RNN model, the same comparison with the real fast insulin absorption curves used by the AIDA2 simulator has been performed. The results are captured in Table 4. The average time to peak found is now 1.65 h, which is a bit higher than the 1.52 h found when using the two-layer RNN. Moving the peak time apart from the bolus injection is consistent with the estimated average value for the decay rate, which is also higher than the decay rate found for the two-layer RNN (0.77 h^−1^ vs. 0.57 h^−1^). The quality of the generated absorption curve for the fast insulin is measured by computing the mean absolute error for the difference between the generated curve and the curve computed by the AIDA2 simulator. Table 4 captures a value of 0.088 for the single layer RNN model which is higher than the 0.077 value in Table 3 for the two-layer RNN model.

Figure 10 shows the real vs. the estimated absorption curves when using the single- layer RNN model using the same patient as in Figure 5. The use of the simpler model is able to find the same peak time, but the values estimated for the decay rate are higher than the value in Figure 5, and the mean absolute error therefore increases.

## 5. Conclusions

This manuscript has presented a method to estimate insulin and carbohydrate absorption curves based on the ability of an RNN machine learning model to achieve optimal predictions for upcoming values of blood glucose levels. Different absorption curves are generated and fed into the RNN in order to select the curves with better blood glucose level predictions over a 60-min horizon. A simple curve generation method using the concatenation of an increasing segment followed by a decreasing segment has been used based on previous studies. The generated curves perform a linear approximation for the increasing blood carbohydrate concentration segment after a meal or the plasma insulin level after a bolus injection. The decaying phase, after the peak value is reached, is approximated using an exponential curve. The peak instant of time and the exponential decay rate are parameters for the curve generation model. The paper has validated the use of a genetic optimization process in order to find the optimal values for the parameters generating the absorption curves. The validation scores for two different RNN models for predicting the blood glucose level in a 60-min prediction horizon have been used as the fitness function for the genetic optimization.

Using a two-layer LSTM-based RNN model provides a better estimation for the fast insulin absorption curves than using a single-layer RNN model. The optimal curves for fast-acting insulin found by the genetic algorithm when using a two-layer RNN are able to estimate the peak instant of time with less than 10 min error. The mean absolute error between the curves provided by the AIDA2 simulator and the optimal curves found by the genetic algorithm is below 0.1 for curves normalized so that the maximum value is 1. Using the two-layer RNN provides an average mean absolute error of 0.077, while the single layer RNN model generates an average mean absolute error of 0.088. 

This manuscript has used the data produced by the AIDA2 simulator, since the simulator provides information about the plasma insulin levels computed for each scenario. These plasma insulin curves made available by the simulator have been used in order to validate the results obtained by the proposed method in this manuscript. As a future study, a real patient dataset will be used in order to assess the improvement in blood glucose estimations when using a prior computation for carbohydrate and insulin absorption curves as proposed in this manuscript.

## Figures and Tables

**Figure 1 sensors-21-04926-f001:**
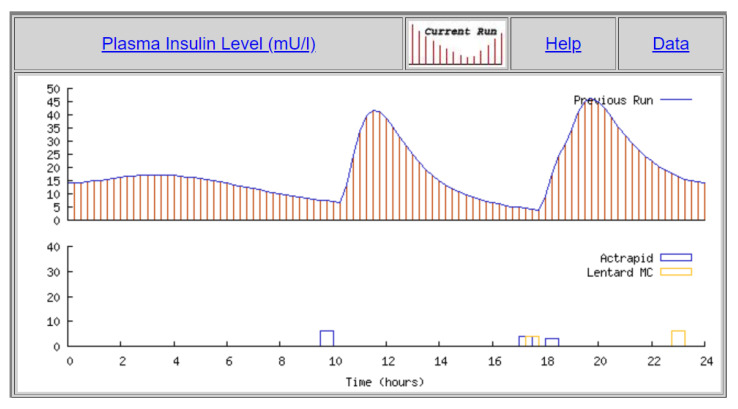
Plasma insulin for one simulation (plot generated by the AIDA2 simulator).

**Figure 2 sensors-21-04926-f002:**
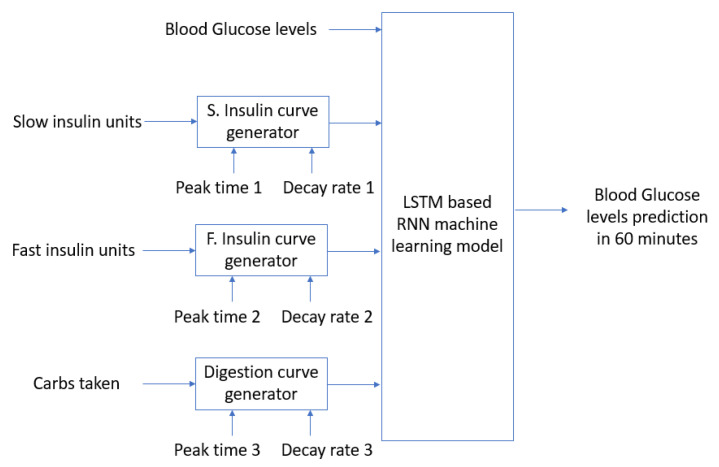
Machine learning model based of insulin absorption and meal digestion curves.

**Figure 3 sensors-21-04926-f003:**
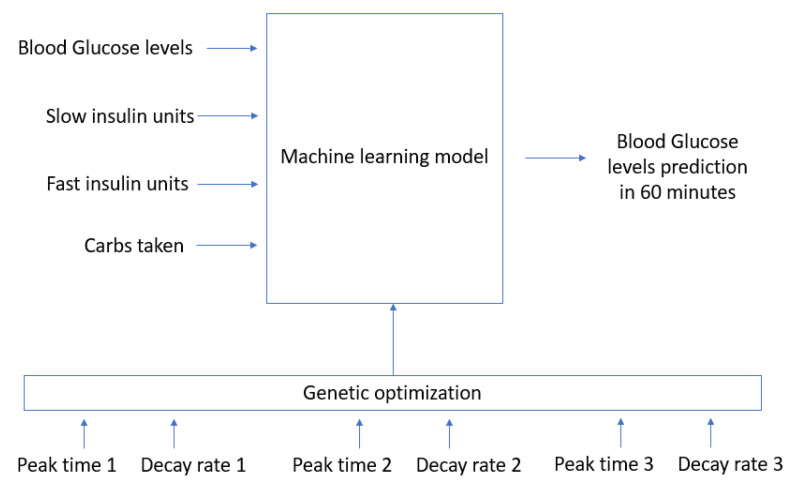
Genetic optimization to find the optimal values for the generated curves.

**Figure 4 sensors-21-04926-f004:**
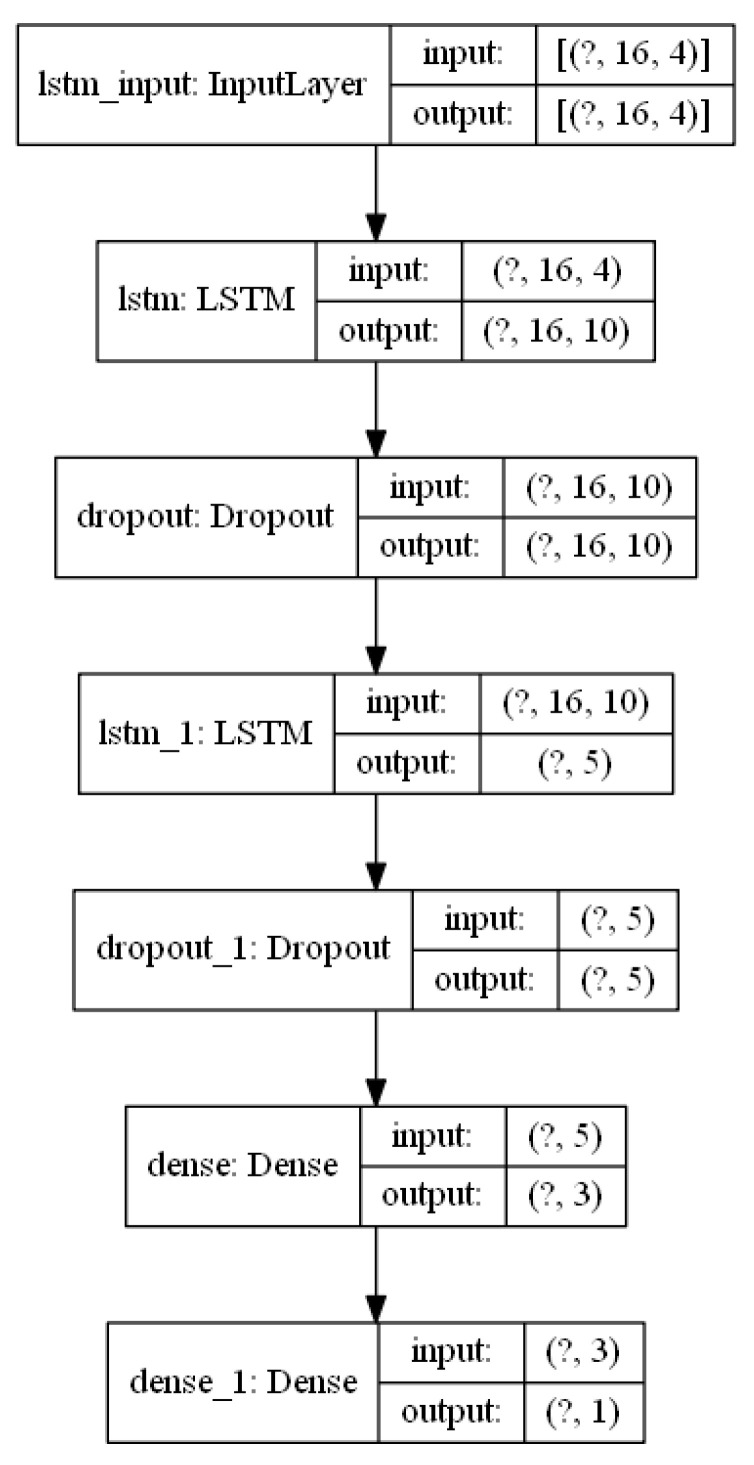
RNN model parameters.

**Figure 5 sensors-21-04926-f005:**
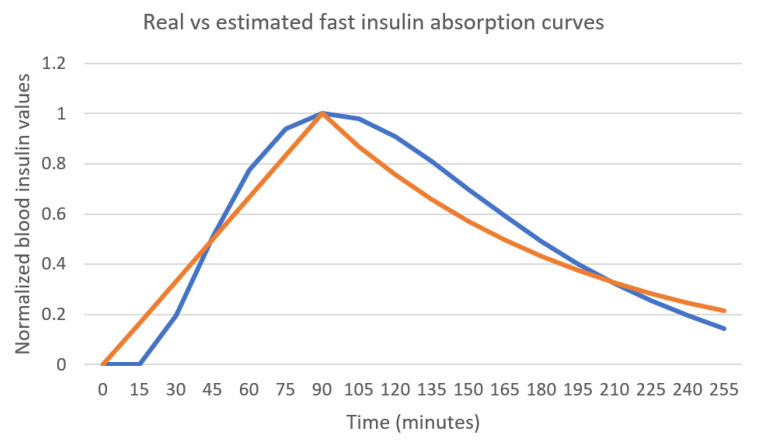
Real vs. estimated fast insulin curves.

**Figure 6 sensors-21-04926-f006:**
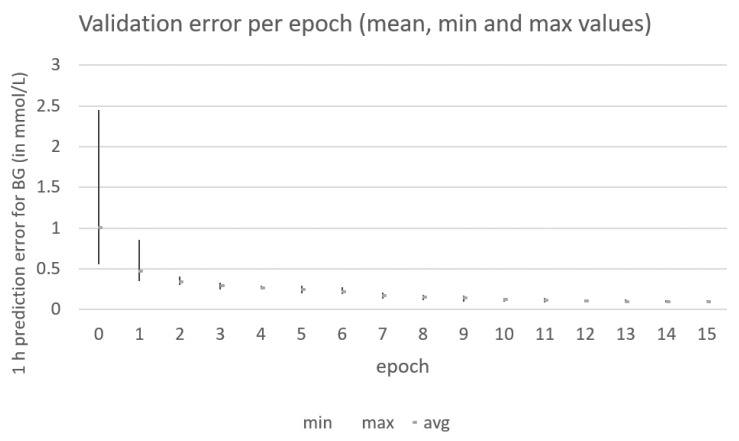
Mean, min, and max prediction errors for the individuals computed at each epoch by the genetic algorithm for one patient generated by the AIDA2 simulator.

**Figure 7 sensors-21-04926-f007:**
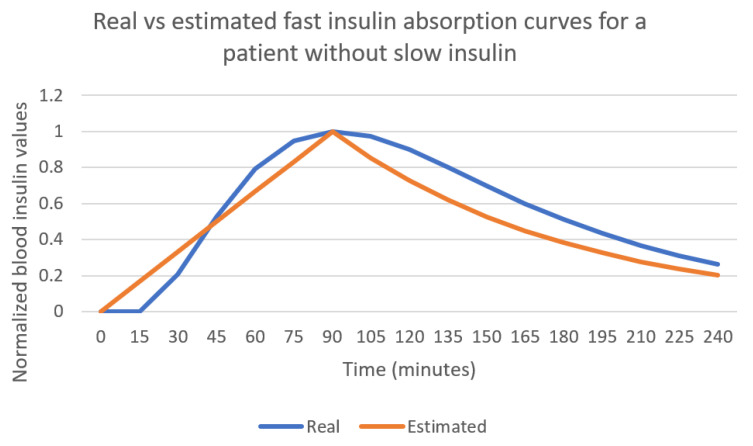
Real vs. estimated fast insulin curves for a patient that does not use slow insulin.

**Figure 8 sensors-21-04926-f008:**
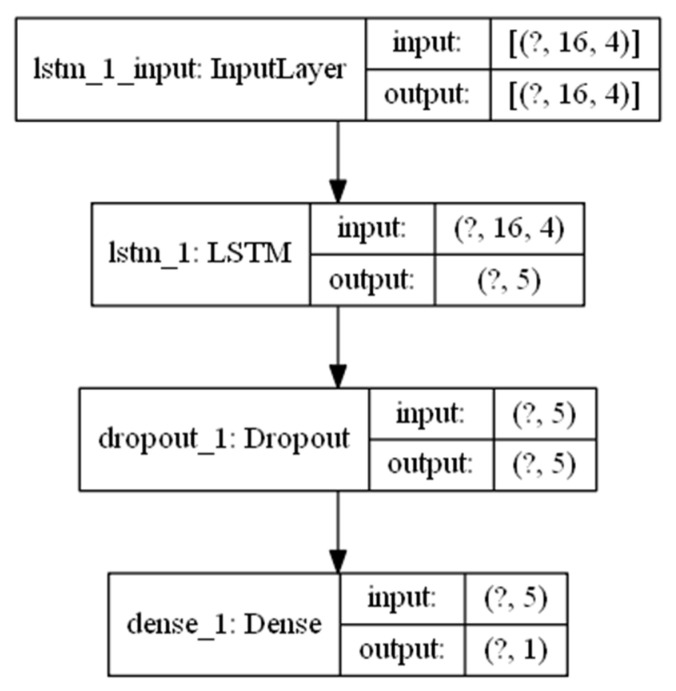
Single LSTM layer RNN model parameters.

**Figure 9 sensors-21-04926-f009:**
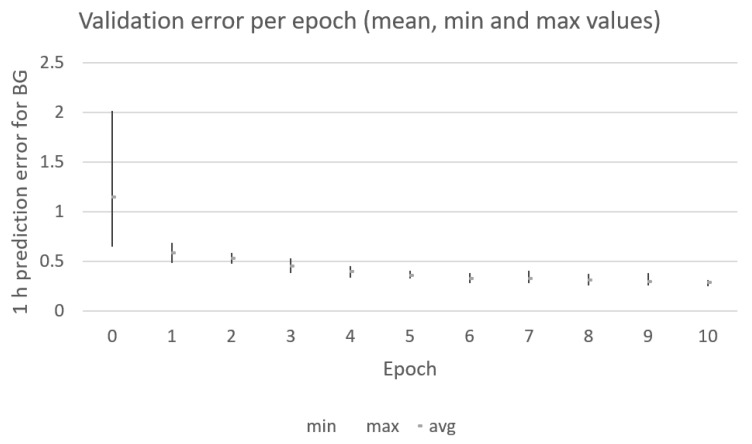
Mean, min, and max prediction errors for the individuals computed at each epoch by the genetic algorithm for the same patient in Figure 5 generated by the AIDA2 simulator.

**Figure 10 sensors-21-04926-f010:**
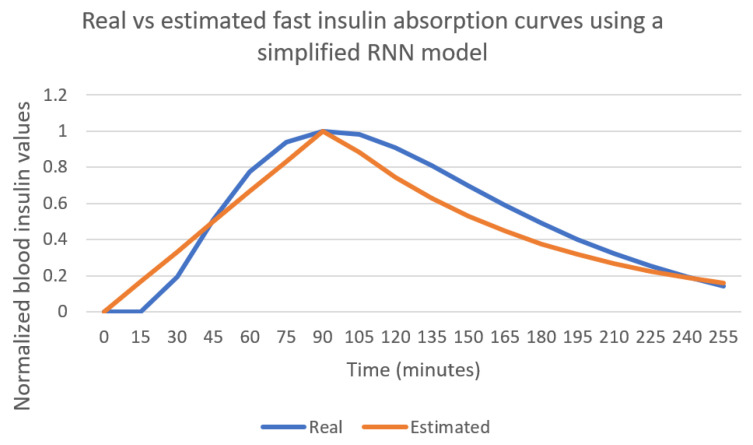
Real vs. estimated fast insulin curves using a single LSTM layer RNN.

**Table 1 sensors-21-04926-t001:** Details for the dataset.

Parameter	Value
Number of simulated patients	40
Number of days per patient	8
Average number of insulin boluses per day	4
Average number of meals per day	6

**Table 2 sensors-21-04926-t002:** RNN model details.

Layer (Type)	Output Shape	#Param
lstm_1 (LSTM)	(None, 16, 10)	600
dropout (Dropout)	(None, 16, 10)	0
lstm_2 (LSTM)	(None, 5)	320
dropout_1 (Dropout)	(None, 5)	0
dense (Dense)	(None, 3)	18
dense_1 (Dense)	(None, 1)	4

**Table 3 sensors-21-04926-t003:** Average results.

Parameter	Value
Mean peak time (h)	1.52
Mean decay rate (h^−1^)	0.57
Mean absolute error	0.078

**Table 4 sensors-21-04926-t004:** Average results.

Parameter	Value
Mean peak time (h)	1.65
Mean decay rate (h^−1^)	0.77
Mean absolute error	0.088

## Data Availability

The data has been obtained from the AIDA2 simulator [12].

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
