# Peer review of "Learning Carbohydrate Digestion and Insulin Absorption Curves Using Blood Glucose Level Prediction and Deep Learning Models"

_sensors, 2021, doi:10.3390/s21144926_

Round 1

Reviewer 1 Report

The authors proposed a deep learning-based method based on the estimation of carbohydrate digestion and insulin absorption curves for blood glucose prediction. This manuscript was designed well and written clearly, the study is novel and interesting. However, there are some aspects that have not been sufficiently explained or justified in this manuscript, that should be reviewed, as specified below.

  • The characteristics of the dataset are not described well. Are there any preprocessing techniques applied to the experimental data before learning?
  • The genetic algorithm used in this study should be explained in detail;
  • There are not enough comparative analyses with other learning algorithms (GRU, MLP, SVM, etc). It is not clear to capture the efficiency of the proposed method in both computation and performance levels;

Author Response

All the comments from the reviewers have been fully considered and implemented in order to improve the current version of the manuscript. We want to thank the reviewers for their feedback and comments that have allowed us to improve the quality of the paper.

Answer to the comments:

Comment 1: The characteristics of the dataset are not described well. Are there any preprocessing techniques applied to the experimental data before learning?

Answer:

Thank you for the suggestion since it is true that more details are very convenient to describe the dataset. We have added the following information to section 3.1:

The characteristics of the dataset used are captured in Table 1. For each case scenario provided by the AIDA 2 simulator [10] a total of 8 days of data is generated. The average number of insulin injections per day has been set to 4 and the average number of meals per day has been set to 6.

Table 1. Details for the dataset.

Parameter

Value

Number of simulated patients

40           

Number of days per patient

8           

Average number of insulin boluses per day

4                

Average number of meals per day

6                

More information about the preprocessing techniques used has also been added:

In order to feed the model in Figure 2, the input data is pre-processed in order to generate 4-hour windows of data for each of the input signals. No further pre-processing steps have been required since the data contains no measurement errors.  

Comment 2: The genetic algorithm used in this study should be explained in detail;

Answer:

 The genetic algorithm used in this study is now explained in detail in section 3.2:

A genetic algorithm is used in order to find the optimal values for the 6 parameters controlling the generation of the absorption curves. In particular, the algorithm described in chapter 7 in reference [24] is used. The algorithm concatenates 4 major functions in order to find the individuals that optimize the value of a fitness function: recombine, mutate, evaluate and select. The recombine function generates individuals which combine the attributes from those in the previous generation in order to find new individuals. The mutate function makes some random changes with a particular probability to the attributes in each individual. The evaluate function calculates the fitness values for all the individuals in a population. Finally, the selection function chooses the best individuals in order to generate a new population. The algorithm can iterate several times or epochs. An initial population of 15 individuals has been used, including a mutation probability of 20% in order to generate 15 generations of individuals. The best-fitted individuals have been selected for each generation. The process is shown in Figure 3. The initial values for each individual have been randomly selected. The maximum and minimum values for peak times have been set from 0 to 2 hours for meal digestion and fast insulin concentration and from 0 to 12 hours for slow insulin absorption

Comment 3: There are not enough comparative analyses with other learning algorithms (GRU, MLP, SVM, etc). It is not clear to capture the efficiency of the proposed method in both computation and performance levels;

Answer:

A comparative analysis with a previous study using the same dataset but with no absorption curves is now added to the manuscript (in section 4.3). The following text has been added:

Comparing the prediction results with those in Table 4 in reference [21], the addition of absorption curves fitted to each patient, as proposed in this paper, is able to improve the Root Mean Square Error (RMSE) when using similar LSTM based deep learning models. The authors in [21] achieved a RMSE value of 3.45 mg/dL using the same dataset as in this paper but only for a prediction horizon of 30 minutes and an optimized deep learning model. Using the two-layer model in section 4.1, the absorption curves are able to improve the RMSE value to 1.71 mg/dL for a prediction horizon of 60 minutes.   

The table mentioned in the text (Table 4 in reference [21]) contains a comparison with previous methods as the reviewer suggests and we have therefore only referenced it so that we do not repeat all the information in the current manuscript.

Reviewer 2 Report

The submitted manuscript applies a deep learning algorithm for absorption curves. It is well written but could be accompanied by additional technical information.
Comments:
1.    Sect. 3.1: It could be valuable to include technical details of the mathematical model used to simulate the data with the AIDA2 simulator. These details could also be moved to an appendix. 
2.    Sect. 3.1: I am wondering whether the proposed modeling strategy allows heterogeneity between persons? Has the model to be trained separately for each subject?
3.    L. 204: please provide references for recurrent neural networks (RNN).
4.    Table 1, p. 6: rename column into ”#Param“
5.    Fig. 4, p. 7: Is this figure self-explaining without providing details of the RNN algorithm?
6.    Tables 2 and 3 only contain three numbers. Consider moving them in the text because such small tables are probably not needed.
7.    Fig. 6/Fig. 9: Missing symbols for labels ”min“ and ”max“?
8.    Fig. 7/Fig. 10: missing labels for lines.
9.    L. 313: typo ”de“
10.    I find figures produced in Excel (or similar software) not appropriate for scientific publications. Consider a more suitable software for creating figures.
11.    A list of abbreviations is missing.

Author Response

All the comments from the reviewers have been fully considered and implemented in order to improve the current version of the manuscript. We want to thank the reviewers for their feedback and comments that have allowed us to improve the quality of the paper.

Answer to Comments:
Comment 1.    Sect. 3.1: It could be valuable to include technical details of the mathematical model used to simulate the data with the AIDA2 simulator. These details could also be moved to an appendix. 

Answer:

The following information has been added to section 3.1:

The model is based on 4 differential equations along with 12 auxiliary relations which are captured in [10]. The derivative of the insulin concentration over time has a positive term proportional to the rate of insulin absorption divided by the volume of insulin and a negative term controlled by the rate of insulin elimination [10]. The change of glucose concentration with time is also guided by a differential equation which takes into account the systemic appearance of glucose via glucose absorption from the gut, the overall rate of peripheral and insulin-independent glucose utilization, the net hepatic glucose balance, the rate of renal glucose excretion and the volume of distribution for glucose [10].

Comment 2.    Sect. 3.1: I am wondering whether the proposed modeling strategy allows heterogeneity between persons? Has the model to be trained separately for each subject?

Answer:

 Thank you for the comment since it is very relevant. The model is fit to each patient. Previous studies such as [21] has shown that training the model with data of several patients provides worse results. The following information is captured in section 4.2:

The parameters defining the differential equations implemented by the AIDA2 simulator [10] for each patient consider different characteristics which try to accommodate real cases for different users. The model in Figures 2 and 3 has therefore been used for each patient independently.

Comment 3.    L. 204: please provide references for recurrent neural networks (RNN).

Answer:

Reference [19] has been added.

Comment 4.    Table 1, p. 6: rename column into ”#Param“

Answer:

Done

Comment 5.    Fig. 4, p. 7: Is this figure self-explaining without providing details of the RNN algorithm?

Answer:

A reference has been provided to a similar structure in [19]. The Layers in Fig. 4 are self-explaining and Fig. 4 provides the information about how these layers are concatenated and the number of hidden units and memory cells so that the architecture can be easily replicated.

Comment 6.    Tables 2 and 3 only contain three numbers. Consider moving them in the text because such small tables are probably not needed.

Answer:

We have tried to remove the tables and incorporate the information in the text but we think it gets less clear. We have however converted the table so that now it contains several rows which may be better for clarity.

Comment 7.    Fig. 6/Fig. 9: Missing symbols for labels ”min“ and ”max“?

Answer:

Thank you for the comment. The upper part of the vertical line corresponds to the maximum value and the lower part to the minimum value. This is why no especial symbol is used in the legend.

Comment 8.    Fig. 7/Fig. 10: missing labels for lines.

Answer:

A legend has been added to both figures.

Comment 9.    L. 313: typo ”de“

Answer:

Done

Comment 10.    I find figures produced in Excel (or similar software) not appropriate for scientific publications. Consider a more suitable software for creating figures.

Answer:

Thank you very much. We have used Excel in order to get all the results in a homogeneous format but the plots could be generated in Python in the final edition if needed.  

Comment 11.    A list of abbreviations is missing.

Answer:

Thank you for the comment. We have defined each abbreviation the first time that it is used following the layout for MDPI publications:

https://www.mdpi.com/authors/layout

An additional list of abbreviations could be added if needed.  

Round 2

Reviewer 1 Report

Thank you for your modifications. I am satisfied with it.